# Phylodynamics of a regional SARS-CoV-2 rapid spreading event in Colorado in late 2020

Kristen J. Wade[1], Samantha Tisa[1], Chloe Barrington[1], Jesslyn C. Henriksen[1], Kristy R. Crooks[2], Christopher R. Gignoux[2], Austin T. Almand[3¤], J. Jordan Steel[3], John C. Sitko[3], Joseph W. Rohrer[2], Douglas P. Wickert[3], Erin A. Almand[3], David D. Pollock[1]*, Olivia S. Rissland[1]*

**1** Department of Biochemistry and Molecular Genetics, University of Colorado School of Medicine, Aurora, Colorado, United States of America, **2** Colorado Center for Personalized Medicine, University of Colorado School of Medicine, Aurora, Colorado, United States of America, **3** Department of Biology, United States Air Force, Colorado Springs, Colorado, United States of America

¤ Current address: University of Colorado School of Medicine, Colorado Springs, Colorado, United States of America

* david.pollock@cuanschutz.edu (DDP); olivia.rissland@gmail.com (OSR)

**Data Availability Statement:** All genome sequences were deposited in the GISAID database, accession numbers available in S1 File. The full output and tree statistics from the BEAST analysis

## Abstract

Since the initial reported discovery of SARS-CoV-2 in late 2019, genomic surveillance has been an important tool to understand its transmission and evolution. Here, we sought to describe the underlying regional phylodynamics before and during a rapid spreading event that was documented by surveillance protocols of the United States Air Force Academy (USAFA) in late October-November of 2020. We used replicate long-read sequencing on Colorado SARS-CoV-2 genomes collected July through November 2020 at the University of Colorado Anschutz Medical campus in Aurora and the United States Air Force Academy in Colorado Springs. Replicate sequencing allowed rigorous validation of variation and placement in a phylogenetic relatedness network. We focus on describing the phylodynamics of a lineage that likely originated in the local Colorado Springs community and expanded rapidly over the course of two months in an outbreak within the well-controlled environment of the United States Air Force Academy. Divergence estimates from sampling dates indicate that the SARS-CoV-2 lineage associated with this rapid expansion event originated in late October 2020. These results are in agreement with transmission pathways inferred by the United States Air Force Academy, and provide a window into the evolutionary process and transmission dynamics of a potentially dangerous but ultimately contained variant.

## Introduction

The COVID-19 pandemic caused by SARS-CoV-2 has resulted in worldwide disruption and more than 6.4 million recorded deaths [Worldometers, July 28, 2022]. Sequencing the SARS-CoV-2 genome from infected individuals is an effective means to track the dispersion and prevalence of the virus. Genomic tracking allows researchers to model viral evolution and identify possible variants of concern. For example, sequencing provided strong evidence that

are available in Supporting Information S1 Dataset. All variant information and codes/script written specifically for this project are available within the Supporting Information.

**Funding:** KW is supported by NIH R01 GM083127. CB is supported by the T32 grant T32GM136444 awarded to the Molecular Biology graduate program. This work was supported by NIH grants R35GM128680 (OSR) and the RNA Bioscience Initiative. The funders had no role in study design, data collection and analysis, decision to publish, or preparation of the manuscript.

**Competing interests:** The authors have declared that no competing interests exist.

the virus was transmitted locally within Washington state as early as January 2020 [1]. Similarly, sequencing allowed public health officials to track the rise and spread of the highly infectious Delta variant, enabling more responsive policies [2]. These sequencing efforts provide even greater power when coupled with viral evolutionary modelling (phylodynamics) in an epidemiological context [3, 4]. This type of combined approach for tracking and predicting viral transmission is known as genomic surveillance and is a critical component of the modern public health response to viral epidemics [2, 5, 6].

Due to the nature of the pandemic, many grass-roots sequencing efforts sprang up *de novo* around the world, leading to heterogeneity in sequencing quality and inconsistency in geographic sampling. For example, long-read Oxford Nanopore technology (ONT) with overlapping polymerase chain reaction (PCR) amplicons is commonly used to obtain sequences [7]. Long-read ONT has many advantages over short-read sequencing, including that it avoids having to connect short reads that fall away before making it through amplicons and avoids some deterministic errors of other technologies [8], but its high error rate poses other challenges including high false positive and false negative variant calls when comparing to a single reference sequence [8–11]. Because accumulated mutations are key data for inference of phylogenetics, convergence, and selected variants, it is important to be as confident as possible in mutational signatures. There has also been heterogeneity in sequencing between different areas. For instance, analyzing the GISAID dataset [12, 13], there was little sequencing of Colorado genomes for much of 2020, making it challenging to understand the landscape of SARS-CoV-2 variant origins and evolution of local transmission during this phase of the pandemic. Finally, there also are conceptual challenges for this type of genomic surveillance due to incomplete knowledge of the etiology of epidemics, including stochastic environmental effects, sociological response, and phenotypic variance [14].

In the case study presented here, we focus on providing a snapshot of infection dynamics in Colorado in August to November 2020. Sequence data was collected as RNA isolated from infected individuals sampled from two Colorado populations: samples collected for clinical testing at the University of Colorado (CU) Anschutz Medical Campus; and samples gathered predominantly from asymptomatic, randomly sampled individuals at the United States Air Force Academy (USAFA) who tested positive by PCR testing. Together, these samples represent a baseline for the two largest Colorado cities (Denver/Aurora and Colorado Springs) in late 2020 and document the rapid initial spread and subsequent containment of a highly-evolved variant. We used replicate sequencing to validate variants across multiple related genomes rather than relying on only a single distant reference sequence to validate variant calls (Fig 1). Having established a credible set of high confidence SARS-CoV-2 genomes, we then describe the phylodynamics of a rapid viral spread event within a relatively controlled environment, estimate its divergence from related samples and compare to its associated epidemiological data [15].

## Materials and methods

### Ethics statement

The United States Air Force Academy Institutional Review Board (IRB) determined the surveillance testing (FAC20200035E) was approved as Not Human Subjects Research in accordance with Title 32, Subtitle A, Chapter I, Subchapter M, Part 219: Protection of Human Subjects, Department of Defense Instruction 3216.02: Protection of Human Subjects and Adherence to Ethical Standards in Department of Defense-Conducted and–Supported Research and Air Force Instruction 40–402: Protection of Human Subjects in Biomedical and Behavioral Research.

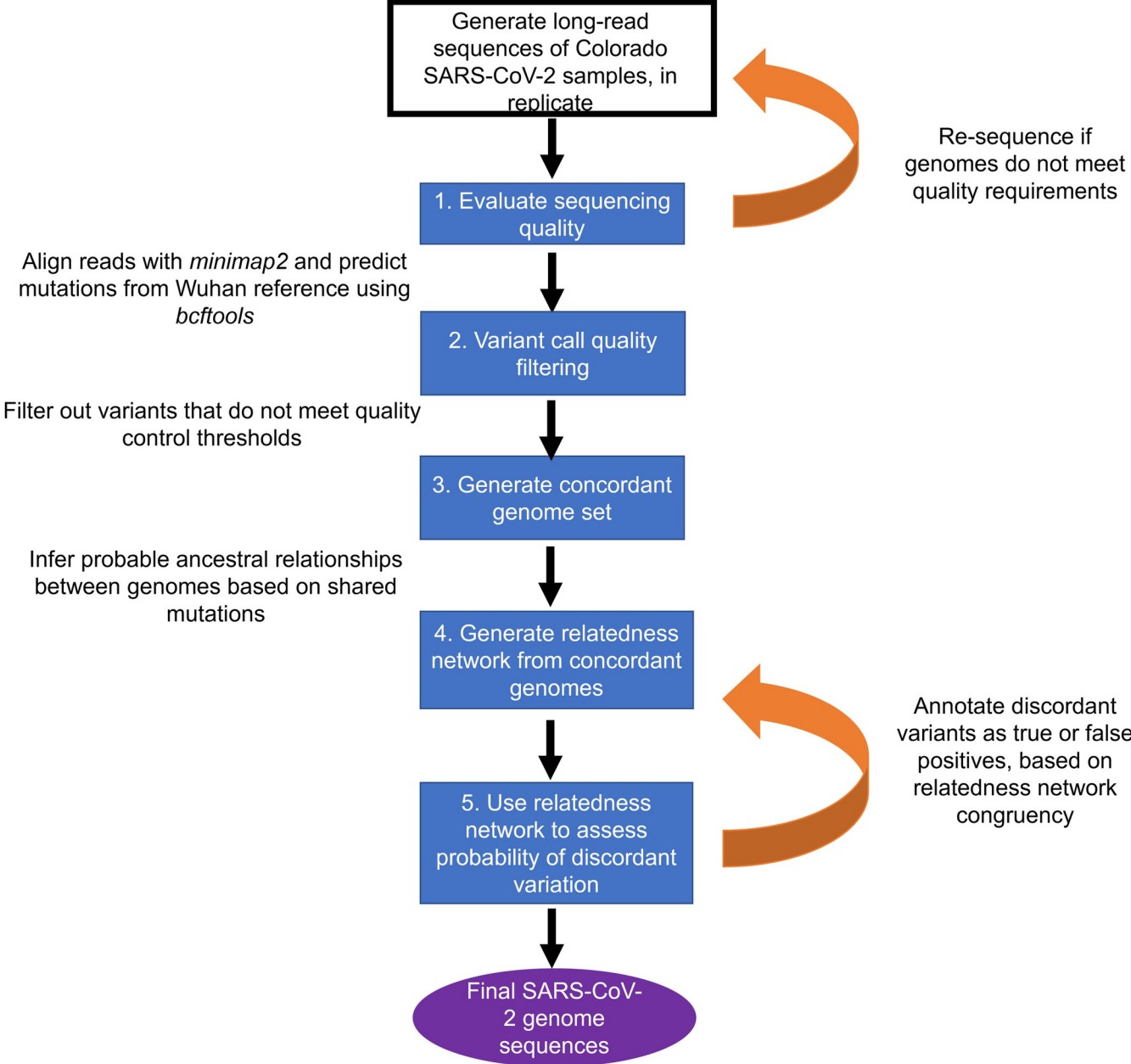

**Fig 1. Flow chart of SARS-CoV-2 sequencing, variant prediction, variant quality analysis and relatedness network generation.** Blue boxes describe the purpose of each stage of analysis, free text designates the specific method used at each stage and orange arrows indicate points where steps were repeated iteratively until sufficient quality metrics were fulfilled.

**Public release.** This manuscript has been approved for public release: PA number: USA-FA-DF-2022-648

**Disclaimers.** The views expressed in this paper are those of the authors and do not necessarily represent the official position or policy of the U.S. Government, the Department of Defense, or the Department of the Air Force.Use of the University of Colorado Biobank samples for this study was reviewed by the COVID-19 Biobanking Committee on May 20, 2020 and approved May 22, 2020 in letter communicated by Matthew J. Steinbeiss, Special Projects

Manager, Office of Regulatory Compliance, University of Colorado Denver, Anschutz Medical Campus, and signed by Thomas Flaig, Vice Chancellor for Research, University of Colorado Denver, Anschutz Medical Campus.

All samples were processed in the Rissland laboratory under IBC#1366. We were advised by Taylor Brumbelow (Human Research Protections, COMIRB Investigator Support, University of Colorado Denver, Anschutz Medical Campus, comirb@ucdenver.edu) on May 19, 2020, that our research qualifies as "non-human subjects research" because we did not have access to or use identifiers such as exact dates of service or admission, county, or zip code.

### Nanopore long-read sequencing

Extracted RNA was obtained from either the University of Colorado BioBank (CU) or USAFA for samples collected between August and November 2020. Sequencing was performed according to the Nanopore Protocol for PCR tiling of SARS-CoV-2 (revision E, released Feb 6 2020) using the V3 primers (https://github.com/artic-network/artic-ncov2019/tree/master/primer_schemes/nCoV-2019/V3). 11 µL RNA was used for reverse transcription and initial amplicon PCR. Samples were processed in random order, and each was sequenced in at least two replicates. After the PCR and bead clean-up, samples were run on a 1.5% agarose gel, and those with visible bands at 400 bp were quantified using a Qubit fluorometer. Samples were end-prepped, barcoded and pooled together for sequencing. Samples were quantified using a Qubit fluorometer. The samples were then processed for downstream sequencing and analysis according to the Nanopore protocol. Sequencing was performed using R9.4.1 (FLO-MIN106D) Nanopore flow cell. Half of the prepared DNA library (7.5 ul) was diluted to a total volume of 12 uL prior to loading. A minimum of 40,000 reads was collected per barcoded sample. Samples with fewer than 40,000 reads were re-sequenced in later runs. Adapter sequences and primers were removed from reads with the Nanopore sequencing software MinKNOW.

Genomes were sequenced in batches. A negative control (water) and a positive control (SARS-CoV-2 RNA from ATCC) were included in each batch of samples. At least two sequencing replicates were performed for each sample to confirm that variants were reproducible; in some cases, especially for low concentration samples, samples were sequenced three or four times to improve call certainty. In later runs, to add further robustness against possible experimental artifacts, the two sequencing replicates were separated, and the order of samples was randomized within a batch.

### Read alignment to the Wuhan reference genome

Following sequencing, reads for each barcode [each barcode corresponding to an individual sample] in fastq format were aligned to the reference Wuhan SARS-CoV-2 genome (NC_045512.2) using *mimimap2* [16]. *Minimap2* parameters were run as follows for each barcode: -a–x splice–uf–k14 –secondary = no NC_045512.2.fasta. The resulting aligned reads were output into sam format and *samtools* [17, 18] was used to generate a binary alignment map (bam). BLAST was performed to verify that primers had been trimmed from read ends. Summary statistics describing read length, error rate, total number of reads, number of mapped reads, and other quality metrics were generated using *samtools stats*. Further sequencing quality assessment was performed with *bedtools genomecov* [19]. For each barcode, a coverage histogram (-ibam), coverage map across consecutive intervals (-ibam–bga) and coverage map at each single nucleotide (-ibam–d) were generated. Finally, an R package (*minionCovidCoverage.R*) was used to visualize the genome-wide coverage distribution for each barcode individually. A custom python script (*findLowQualBases.py*) was used to obtain coordinates of low-quality genome sequence and stored in a bed file to mask the low-quality regions from the

consensus sequence at a later step. Steps described here were called from a pipeline wrapper script, *runCovidSeqs.sh (Code available at: https://bitbucket.org/pollocklaboratory/ covid19phylodynamicscode2021/).*

## Sequencing quality assessment

Using the aforementioned coverage and read information, each individual genome included in further analysis was required to meet a series of sequencing quality thresholds.

Each successfully sequenced genome had:

1. A total read count greater than or equal to 40,000 reads.

2. An error rate less than or equal to 11%

3. Greater than 95% of reads mapped to the reference genome

4. Low quality sequence content less than 5% of the genome length

5. Low-coverage ($< = 30X$) sequence at the 5' and 3' end of the genome that did not exceed two hundred nucleotides in length on each end.

Any samples that failed to meet these thresholds were re-sequenced. If genome quality did not meet the above standards upon resequencing, the genome was excluded from analysis.

## Prediction of mutations from the reference genome

Variant likelihoods at each position were generated from the bam files of high-quality samples using the mpileup package of bcftools-1.11 [18], with the following parameters: -oU–d 200000. We then used bcftools call to make the variant calls, with respect to the reference genome (NC_045512.2), under the following setting:—ploidy 1 –vm–Oz. Variants assigned a 'QUAL' quality score <50 (ranging from 0–225) were excluded, but stored separately should revision be required. Variants were stored in variant call format (vcf) and variant calls were mapped to the reference genome sequence (NC_045512.2) with bcftools consensus. Following previous publications [20, 21], positions with coverage less than or equal to 30X were masked from each genome, or marked as 'N' in the consensus genome sequence using bedtools maskfasta. Therefore, no variants were called at these regions. This process generated a putative consensus sequence for each SARS-CoV-2 sample replicate individually, which represented its predicted mutations from the Wuhan reference sequence.

## Additional quality control and generation of concordant genome set

Following the variant prediction process, we further evaluated each replicate's variant predictions for the following quality criteria:

- If a predicted variant position fell within 200 bases from either the 5' or the 3' end of the genome, it was excluded due to consistently poor coverage in those regions.

- Any variants that occurred in stretches of low-quality score, extending across more than one adjacent or nearby position, were excluded as a likely artifact.

- Putative mutation events at 28881–28883 were excluded, as this region is known to frequently recur as post in-situ (PCR or sequencing) [22–26]

Following these additional quality control steps, we evaluated each genome's set of replicates for consistency in variant calls. Genomes in which all replicates were predicted with the same set of variants were considered concordant. However, there were instances of

inconsistent variant calls between replicates of the same SARS-CoV-2 genome. This subset was described as discordant and set aside for later evaluation. This was done as suggested by Robasky et al., 2014.

## Assessment of possible primer match-derived artifacts

To evaluate if primer error had contributed to recurring variant artifacts, primer sequences were mapped to all reads of a given barcode. The position of the primer match, relative to each read, was tracked and a distribution was created of all primer match positions across all reads (S1 Fig). The average matching read position of each primer was evaluated to assess whether any primers were enriched for mapping to ends of reads, or were distributed randomly as expected.

## Inference of SARS-CoV-2 ancestral relatedness network

Plausible common ancestors for all 28 concordant genomes in the CU and USAFA sequences were independently inferred based on mutations in our dataset that deviate from the reference genome NC_045512.2. To better understand the larger, continental context of our sampling of genomes, plausible ancestors of the CU and USAFA genomes were qualitatively compared to the NextStrain "ncov" database sample of 3983 genomes accessed on May 11, 2021 [22]. Of these, 3923 were submitted in North America between March 5, 2020 and May 10, 2021, and will be referred to as North American NextStrain (NANS). For each mutation event in our newly inferred genomes, the estimated NANS frequency was considered at four different time points: August 14, 2020; November 14, 2020; February 13, 2021; and May 5, 2021. Each mutation event was also associated with one of the eleven major NextStrain clades based on the ancestral context in which it first appeared. This was done to provide a reference point for understanding where the Colorado genomes fit in context, relative to the viral dynamics across the continent at the time.

Although NextStrain calls its groupings "clades", meaning they are collections of all viruses inferred to be descended from a single common ancestor, their tracking and labelling system confusingly goes against common usage of the word 'clade' by removing named clades from within larger clades, leading to a paraphyletic naming convention. Their naming conventions also do not track sequential origins of clades within clades, and they do not include many of the CO sequences. We chose to explicitly track and label inferred ancestors and their relationships to CO descendants.

Having obtained a continental, phylogenetic context for where our Colorado samples likely originate, we then inferred ancestral lineages and relatedness within our SARS-CoV-2 dataset. Nodes connecting each CU and USAFA genome sequence to their most plausible direct ancestor were assigned by grouping shared sets of mutations between our genomes. Plausible ancestral genomes were reverse-inferred in a parsimonious approach by working backwards from our genome dataset, and assigning mutations to ancestral nodes based on shared sets of mutations. Due to the rarity of variants compared to the length of the genome and the short time range of the closely related sequences in this study, back mutations and multiple mutation events at a single site are not likely and were not considered. Branching events separating these inferred, plausible ancestors were provisionally assigned based on differences in mutation content. These plausible common ancestors represent plausible branching points at which novel mutations appeared in the genomes included in our dataset. Non-ancestral mutation events were inferred if they occurred in only one genome across our sampled dataset and did not correspond to any known mutation already in the NANS database. These were assigned to the branch leading to the individual sequences in which they occurred. This process was used to

infer a plausible ancestral relatedness network among SARS-CoV-2 samples in our dataset, and are used to organize discussion of these sequences. These plausible ancestral nodes were labelled A1-A13.

## Use of relatedness network to resolve incongruent relatedness

The approximate relatedness among genomes was used to interpret variant calls in the discordant variant set from above, noting that all such variants were generally rare or absent from the worldwide sequences available at that time. Each genome was fitted to the existing network, based on its variant set with and without the discordant variants. We first considered that a variant that was discordant among replicates of a sample but had no effect on the structure of the network was considered a valid variant call. Most cases were of this nature because most discordant variants were supported by multiple calls in related genomes, and were presumed to be false negatives in some replicates. We disallowed any discordant variant that altered the relatedness network, conservatively presuming that it was a possible false positive variant and not well supported. We discuss removal of variants in further detail in the results. We ended with a complete set of confident variants for each genome that resolved discordancies among replicates so that they did not impact inferred relatedness networks among the SARS-CoV-2 genomes in our dataset and had no impact on our results. Finalized genomes were aligned with Mugsy v1r2.2 [27]. IQTree web server [28] was then used to generate a phylogenetic tree, using the settings: -s covidGenomes22Ref.fasta -st DNA -m JC+F -bb 1000 -alrt 1000 -o NC_045512.2. This tree was then visualized using FigTree v1.4.4.

## Estimation of CO Springs lineage divergence time

The Bayesian Evolutionary Analysis Sampling Trees (BEAST) suite, version 1.10.4, was used to generate evolutionary divergence estimates from the USAFA SARS-CoV-2 genomes and their respective sampling dates between August 9, 2020 and November 19, 2020 [29–32]. Sampling dates were represented as the number of days since December 26, 2020, or the date associated with the Wuhan reference genome sequence. The evolutionary model was designed in BEAUti, using the following parameters: Exponential growth model, strict clock, HKY substitution model-estimated base frequencies with a 4-category gamma site heterogeneity model. Default priors were used, with the exception of the coalescent population size parameter, which was set to a lognormal distribution, mu = 1, sigma = 10. Operators were set to auto-optimized parameters. The control file containing these model parameters (covidGenomes22Ref_2_allAFonly_2m_exp_model.XML, SDataset 1) was analyzed with BEAST. The MCMC analysis was run for 20 million generations, with a burn-in of 2 million. The resulting posterior distributions were visualized and assessed in Tracer, version 1.7.2. TreeAnnotator was used to summarize the set of BEAST-predicted trees onto a single, target tree. A highest posterior density (HPD) of 95% was chosen for the credible interval of divergence estimates at each node across all trees generated in the analysis. The resulting tree annotation was visualized using FigTree software.

## Comparison to USAFA contact tracing and high-level contact tracing

A large number of the USAFA samples are descended from a series of closely related plausible common ancestors beginning with A12 that are mostly closely related to ancestor A3 (corresponding to NextStrain cluster 20G) but separated by nine mutation events mostly not seen elsewhere in the NextStrain database. This branch of the network appears to have been introduced into USAFA from contact with the local Colorado Springs community [15], and we label it the Colorado Springs variant. The network relationships and identities among

sequences were compared to USAFA contact tracing information (under USAFA IRB FAC20200035E) to examine consistency between our inferred genomic ancestral relationships and what was known about the contact path through the USAFA. The network was also used to examine whether the CO Springs variant arose from multiple community transfers or a single community transfer followed by spreading within the USAFA.

## Results

### Overview of SARS-CoV-2 sequencing and variant calling

We obtained 44 nearly-complete SARS-CoV-2 genomic sequences using RNA collected from anonymized individuals in two Colorado populations. The first was collected in August 2020 by the Colorado Center for Personalized Medicine Biobank at the University of Colorado (CU). The second was collected by the USAFA in September to November 2020. The CU samples were predominantly derived from clinical samples from Denver and Aurora obtained at the University of Colorado hospital, while the USAFA samples were primarily obtained from asymptomatic, randomly sampled cadets. In the case of the USAFA samples, because the testing and quarantine protocols established at the beginning of the school year resulted in a population free from COVID-19 when classes started in August [15], subsequent infections almost certainly originated from contact with the local Colorado Springs community.

We followed the ARTIC protocol to produce overlapping short ~450 bp PCR-amplified segments (amplicons), using Minion sequencing and the Oxford Nanopore Technology (ONT) SARS-CoV-2 pipeline. Differences from the Wuhan reference genome sequence (NC_045512.2) were called as described in the methods to predict mutation events in the CU and USAFA genomes. We attempted to sequence 33 CU samples and 68 USAFA samples. Ten CU and 41 USAFA samples did not amplify well enough to sequence, and five USAFA samples did not sequence well despite amplification. Ultimately, each of the 44 SARS-CoV-2 genomes were fully and reliably sequenced at least twice (S1 and S2 Tables).

### Quality control process filters out likely sequencing artifacts

Prior to evaluating genomes for discordancy, we initially filtered out predicted variants that appeared to be sequencing artifacts, or occurred in regions of low sequence quality, as described in the methods (S3 Table). First, we excluded variants at positions 28882–28883 based on previous knowledge that these were problematic mutations [25], although they would not have had a substantial impact on the final phylogenetic network. Next, a series of polymorphisms from positions 19299 to 19550 were observed in several genomes but were consistently of low quality and not well replicated. Therefore these were excluded, even when the variant met the quality threshold (S3 Table). Similarly, a pair of adjacent mutations at positions 24389–24390 were called in consensus sequences from ten CU and USAFA samples but were consistently replicated in only two samples. A further eight mutations were called in only one replicate per sample, and at generally lower-quality scores. Based on these incompatibilities, we excluded these mutation events from all genomes (S3 Table).

### Discordant variation identified via replicate sequencing

Following quality filtering of variant predictions (S3 Table), the majority of variant calls generated by this analysis were found to be in agreement between replicates (concordant). However, we discovered several instances of incongruent variant calls between replicates of the same genome sample. These were described as discordant, according to Robasky et al., 2014. At least one discordant variant call was observed in 14 of 44 SARS-CoV-2 genomes (S4 Table). In

these instances, certain variants were assigned a high enough likelihood score to pass quality filtering (QUAL>50) in at least one replicate, but were not scored above this threshold consistently across replicates, making them possible false positives. These calls were the results of using standard, widely accepted analysis pipelines and quality filters. This finding further underscores the importance of replicate sequencing protocols, even in the context of long-read sequencing. If a replicate sequencing approach had not been followed, these variants would have been summarily excluded as false positives [11]. Therefore, we investigated further to assess the validity of these variants.

## Relatedness networks can resolve discordant variant calls and address putative false positives

We identified all plausible ancestral sequences in the Colorado phylogenetic network based on all observed different combinations of shared differences from the Wuhan reference sequence. This resulted in 13 inferred ancestors, which we label A1 to A13 for purposes of discussion (Figs 2 and 3, S2 Fig). Using the inferred ancestral relationships between concordant genomes (relatedness network), we then critically evaluated the impact of incorporating each of the discordant genomes individually, based on how it's variant set impacted the structure of phylogenetic relationships between genomes. If the inclusion of the putative false positive variants was in agreement with the shared sets of ancestral mutations in related genomes, the variant was retained. If, however, inclusion of a discordant variant created a set of mutations that were incongruent with the shared ancestral variation of related genomes, the variant was excluded as a false positive. Through this approach, we were able to confidently resolve 16 discordant, putative false positives as confident true positives to include in our genome annotations (S4 Table). In a less rigorous process, these mutations would have been rejected from their respective genomes. As a result of this analysis, only one discordant variant, at position 13094, was excluded as a likely false positive because it occurred sporadically on the relatedness network, suggesting it was not a result of a shared ancestral mutation. Together, these results underscore the importance of replicating sample sequencing to improve confidence in results from this amplicon approach [11].

## Relatedness networks can also reveal false negatives

Comparing shared variant sets between ancestrally related SARS-CoV-2 genomes also uncovered instances where variants that probably should have been present were excluded due to low quality scores across all replicates. Under the original codified criteria for rejecting variants due to low quality scores, genomes Z, R and I had calls at some sites that were in disagreement with the relatedness network, in the sense that there would necessarily had to have been reversions at these sites to make the calls compatible with the network (S5 Table). However, these variants were observed in the original analysis but filtered due to the quality score cutoff. In another instance, genomes AR, AQ and AJ appeared to be missing calls for a mutation that probably should have been present, based on mutation content of closely related genomes of ancestor A3. However, this mutation, at position 27964, was called with very low-quality genotype scores in many other genomes (S5 Table), and we inferred that the site in question was in a genomic region that regularly exhibited low-coverage and low-quality/inconsistent variant calls across most genomes in our sampling. This is consistent with poor amplicon amplification in some of these genomes, and we avoid making claims based on the presence or absence of this variant.

Finally, this approach allowed us to identify two instances of chance congruent mutation events (S6 Table). The mutation at position 14187 in genome J is shared with the descendants of ancestral node A12, although J is a descendant of node A4, which is highly divergent from

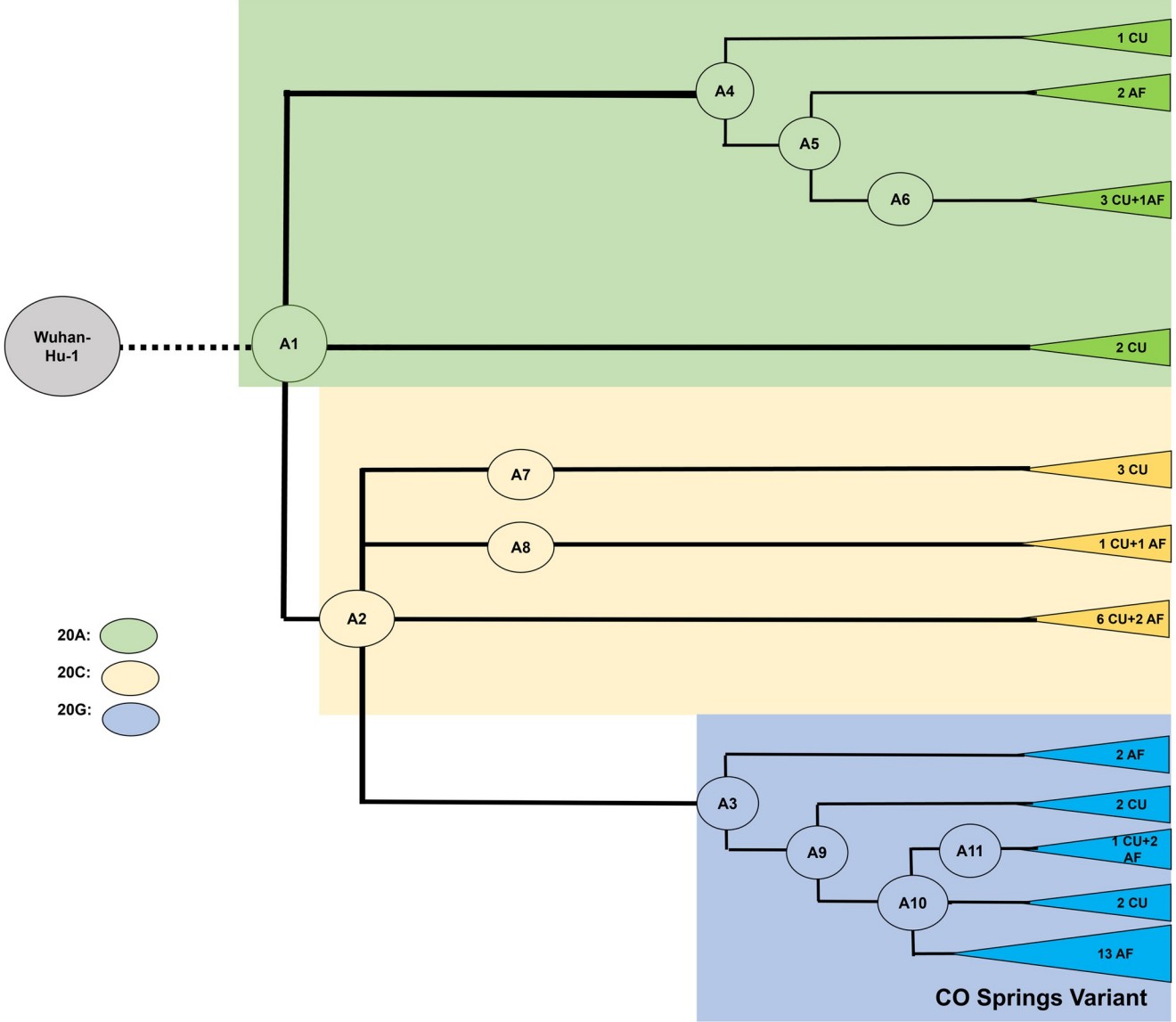

**Fig 2. Major phylogroup clustering of the CUAF SARS-CoV-2 genomes.** Tip labels indicate how many CU Anschutz (CU) and USAFA (AF) genomes are associated with ancestor. Color shading indicates which main phylogroup genomes belong to, A1-A3, corresponding to NextStrain clades 20A, 20C and 20G, respectively.

A12. Additionally, we identified that a mutation at position 24904 occurs in both genomes J and G. However, these genomes are separated by two ancestral nodes, and this mutation is not found in any other related genomes, so we conclude that these two mutations are chance convergent events. The remaining variants are well replicated congruent with the parsimonious phylogenetic network, and represent high-confidence sequences suitable for in-depth evolutionary analysis of SARS-CoV-2 infections in Colorado (S7 Table, S1 Dataset).

## Broad phylogenetic structure of the CU and USAFA genomes

We characterized the ancestral variant composition of each genome, relative to our dataset. We jointly analyzed the entire set of Colorado sequences and placed their plausible common

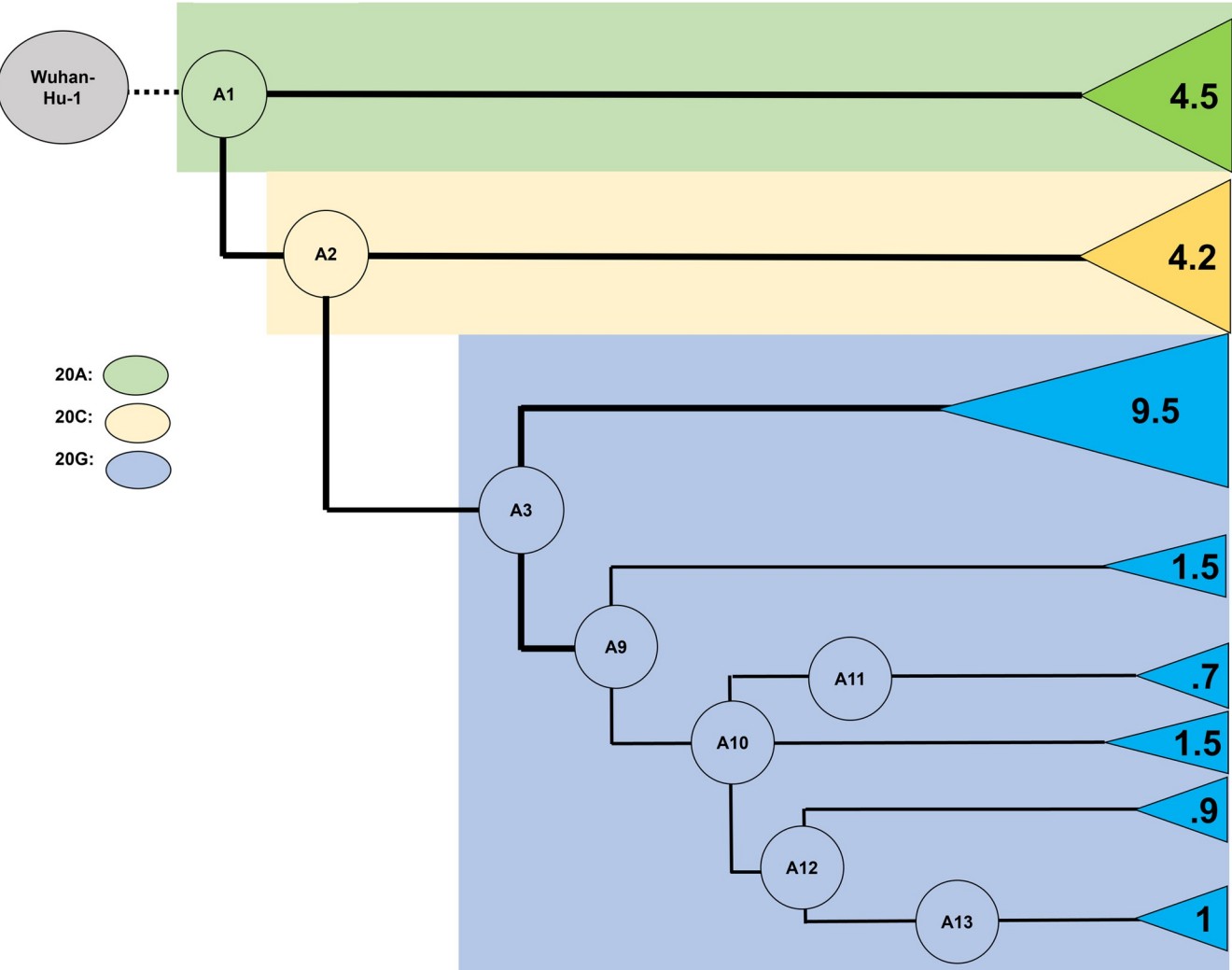

**Fig 3. Closely related subset of Air Force samples suggests a novel, rapidly transmitting strain with low average mutation rate, the "Colorado Springs Variant" (A12-A13).** Numbers at tips indicate average number of mutations per genome in each ancestral grouping.

ancestors on a phylogenetic network. All newly sequenced Colorado genomes appear to be descended from what we label the A1 ancestor, which is the likely ancestor of NextStrain's clade 20A. In other words, they all share four mutation events in common that separate them from the Wuhan reference sequence, at sites 3037, 14408, 241, and 23403. This result is not surprising, as this ancestor contains two amino acid-altering mutations in the RdRp and Spike proteins, which confer a competitive advantage over previous variants since its origin in January 2020 [33]. Based on analysis using NextStrain [22], this variant's early origin and competitive advantage over the original virus caused its descendants to represent 99% of genome sequences throughout the world as of August 2020, when the first of our samples were collected (S3 Fig).

15 of the 21 CU sequences and 75% of the USAFA sequences are descended from A2, which is itself a descendant of A1 and is the likely ancestor of NextStrain's clade 20C, estimated to have originated in April 2020. A2 differs from A1 by two mutations, at sites 25563 and 1059, and rose to a highest frequency of 43% of sequenced genomes in North America by February,

2021. Thus, for some time 20C appeared well on its way to becoming the dominant variant in North America, and may have been at a competitive advantage compared to other early descendants of A1, but has since lost ground to the well-documented Greek-letter variants (Alpha, Beta, Delta, etc.) since then [34–36] (S4 Fig). Because the Colorado A1 and A2 descendants (other than the Colorado Springs variant) are not highly clustered, the higher frequency of A2 in both CU and USAFA samples may indicate that the A2 variant was even more successful in Colorado than in the rest of North America.

In our network, we define all differences that can be parsimoniously mapped to branches of the network to be inferred lineage-defining mutations. There were 41 lineage-defining mutations and 135 novel mutations that were confidently identified from the network of 44 genomes that were sequenced (Table 1, S7 Table). Of the 41 lineage-defining mutations, 24 of these occurred in the ORF1AB gene, five in the S gene, five in the ORF3A gene, one in the ORF8 gene, and six in the N gene. 18 have documented amino acid substitutions already in the NextStrain database.

These mutations were qualitatively cross referenced against the NextStrain ncov database North American frequencies (henceforth, ncov) to evaluate their mutation frequencies and determine which of the eleven major clades, as defined by NextStrain at the time of evaluation, were represented in our sampling (S4 Fig, Table 1). 18 mutations had no detectable frequency in ncov (<<1% in Table 1), and seven mutations were found sporadically across the ncov tree, with a prevalence of ~1%.

Of the 18 Colorado lineage-defining mutations that were more common in ncov, all were on the lineages leading to the series of ancestors A1, A2, or A3, corresponding to NextStrain clades 20A, 20C, and 20G. Based on shared mutations, 22 of the 44 Colorado genomes are descendants of A3 (and thus also A1 and A2), 13 are descendants of A2 (and thus also A1) but not descended from A3, while nine are descended from A1 but not from A2 or A3 (Fig 2). The inferred ancestors other than A1, A2 and A3 are organized such that there are three ancestors (A4-A6) descended from A1 but not A2 or A3, two ancestors (A7-A8) descended from A2 but not A3, and five ancestors (A9-A13) descended from A3. We further verified our relatedness networks by using IQTree to generate a phylogenetic tree of all 44 final, concordant genomes, including the Wuhan-Hu-1 genome, which served as the reference for variant calling (S5 Fig). This confirms our grouping of genomes into the aforementioned relatedness network substructure. In this way, we were able to describe the patterns of relatedness and evolutionary dynamics between 44 Colorado SARS-CoV-2 genomes.

## Origins of a novel lineage associated with a rapid transmission event

Although many of the USAFA samples were likely picked up from the local Colorado Springs community based on detailed contact-tracing information and limits on off-base travel implemented by USAFA, a group of samples were also associated with a rapid-spreading event within the USAFA campus starting in late October 2020 [15]. In looking at the distribution of inferred mutations on the Colorado phylogenetic network, we found that most common ancestors [A4, A5, A6, A8, A9] share a shallow network of divergence from the ancestors A1, A2, and A3, with one or two mutations separating each from an earlier ancestor (Fig 2, S2 Fig). Divergence from ancestors A1 and A2 involved an average of 4.55 and 4.23 mutations/genome (s.d. 2.69 and 3.36, range 1–9 and 0–11), respectively (Fig 3, S8 Table). Among the sampled sequences in this part of the tree, only two are identical. These results are in rough agreement with the idea that the mutations and most of the network diversification (other than A1, A2, and A3) were unselected and that mutations accumulated randomly with a rate for beta coronaviruses between $1.3 \times 10^{-4} – 6.1 \times 10^{-4}$ mutations per site per year [37–40]. A3, which

**Table 1. Lineage-defining mutations in a set of 44 Colorado SARS-CoV-2 genomes.**

| Position[1] | ORF | REF[2] | ALT[2] | Amino acid change[3] | Mutation frequency in US [NextStrain] | | | | Ancestor introduced[4] | NextStrain clade |
|---|---|---|---|---|---|---|---|---|---|---|
| | | | | | August 2020 [8–14] | November 2020 [11–14] | February 2021 [2–13] | May 2021 [5–5] | | |
| 241 | ORF1A | C | T | | <<1% | <<1% | <<1% | <<1% | A1 | 20A |
| 829 | ORF1A | C | T | | <<1% | <<1% | <<1% | <<1% | A13 | NA |
| 1059* | ORF1A | C | T | T265I | 25% | 36% | 27% | 12% | A2 | 20C |
| 1927 | ORF1A | T | C | | <<1% | <<1% | <<1% | <<1% | A11 | NA |
| 2668 | ORF1A | C | T | | <<1% | <<1% | <<1% | <<1% | A7 | NA |
| 3037 | ORF1A | C | T | | <<1% | <<1% | <<1% | <<1% | A1 | 20A |
| 4021 | ORF1A | C | T | | <<1% | <<1% | <<1% | <<1% | A12 | NA |
| 7006 | ORF1A | C | T | | <<1% | <<1% | <<1% | <<1% | A8 | NA |
| 7086 | ORF1A | C | T | | <<1% | <<1% | <<1% | <<1% | A12 | sporadic[5] |
| 10319* | ORF1A | C | T | L3352F | 5% | 26% | 11% | <1% | A3 | 20G |
| 11824 | ORF1A | C | T | | <<1% | <<1% | <<1% | <<1% | A9 | NA |
| 12295 | ORF1A | C | T | | <<1% | <<1% | <<1% | <<1% | A12 | NA |
| 13216 | ORF1A | T | C | | <<1% | <<1% | <<1% | <<1% | A12 | NA |
| 14187 | ORF1B | G | A | | <<1% | <<1% | <<1% | <<1% | A12 | NA |
| 14408* | ORF1B | C | T | P314L | 90% | 94% | 91% | 97% | A1 | 20A |
| 15766 | ORF1B | G | T | V767L | 3% | 9% | 2% | <<1% | A11 | 20G |
| 18424* | ORF1B | A | G | N1653D | 3% | 24% | 11% | <1% | A9 | 20G |
| 18486 | ORF1B | C | T | | <<1% | <<1% | <<1% | <<1% | A6 | NA |
| 18538 | ORF1B | G | T | V1691L | <<1% | 1% | 1% | <<1% | A11 | sporadic[5] |
| 19180 | ORF1B | G | T | | <<1% | <<1% | <<1% | <<1% | A8 | sporadic[5] |
| 19891 | ORF1B | G | T | | <<1% | <<1% | <<1% | <<1% | A9 | NA |
| 20268 | ORF1B | A | G | | <<1% | <<1% | <<1% | <<1% | A5 | NA |
| 21304* | ORF1B | C | T | R2613C | 3% | 24% | 10% | <1% | A10 | 20G |
| 21390 | ORF1B | A | G | | <<1% | <<1% | <<1% | <<1% | A12 | NA |
| 21830 | S | G | T | | <<1% | <<1% | <<1% | <<1% | A13 | NA |
| 22162 | S | T | C | | <<1% | <<1% | <<1% | <<1% | A6 | NA |
| 22255 | S | A | T | | <<1% | <<1% | <<1% | <<1% | A11 | NA |
| 22687 | S | C | T | | <<1% | <<1% | <<1% | <<1% | A11 | NA |
| 23403* | S | A | G | D614G | 98% | 98% | 98% | 100% | A1 | 20A |
| 25563* | ORF3A | G | T | Q57H | 31% | 49% | 31% | 13% | A2 | 20C |
| 25593 | ORF3A | G | C | K67N | <<1% | 1% | <<1% | <<1% | A12 | sporadic[5] |
| 25907* | ORF3A | G | T | G172V | 4% | 25% | 11% | <1% | A10 | 20G |
| 25930 | ORF3A | T | C | S180P | <<1% | 1% | 1% | <<1% | A11 | sporadic[5] |
| 26040 | ORF3A | A | T | | <<1% | <<1% | <<1% | <<1% | A9 | sporadic[5] |
| 27964* | ORF8 | C | T | S24L | 6% | 27% | 11% | <1% | A3 | 20G |
| 28472* | N | C | T | P67S | 3% | 24% | 11% | <1% | A10 | 20G |
| 28655 | N | G | T | | <<1% | <<1% | <<1% | <<1% | A7 | NA |
| 28854 | N | C | T | S194L | 16% | 15% | 4% | 2% | A4 | 20A |
| 28869* | N | C | T | P199L | 5% | 28% | 12% | 3% | A10 | 20G |
| 28887 | N | C | T | T205I | 1% | 10% | 17% | 9% | A9 | 21C |

*(Continued)*

**Table 1.** (Continued)

| Position[1] | ORF | REF[2] | ALT[2] | Amino acid change[3] | Mutation frequency in US [NextStrain] | | | | Ancestor introduced[4] | NextStrain clade |
|---|---|---|---|---|---|---|---|---|---|---|
| | | | | | August 2020 [8–14] | November 2020 [11–14] | February 2021 [2–13] | May 2021 [5–5] | | |
| 29439 | N | A | T | Q389H | <1% | <1% | <<1% | <<1% | A11 | sporadic[5] |

* Mutation is found at frequencies > 1% in the ncov North America dataset

[1] positions refer to aligned location in the Wuhan reference genome

[2] REF and ALT refer to the nucleotide state at that position in the reference genome and the corresponding alternative allele at that position in comparison genomes, respectively.

[3] Amino acid replacement standard notation shows the reference amino acid, the amino acid position in the corresponding gene, and then the inferred amino acid in the alternative genomes, all as determined from ncov.

[4] The clade corresponding to an ancestral sequence was determined as the named clade in the NextStrain database at the time of evaluation with the most mutations shared with the "ancestor introduced" prior to that clade on the NextStrain phylogenetic tree.

[5] Mutation is not associated with a phylogenetic grouping, but occurs at seemingly random branches throughout the entire tree date accessed: 7/12/21

Number of genomes: Showing 4000 of 4043 genomes sampled between Mar 2020 and June 2021

corresponds to the common ancestor of NextStrain's 20G clade, increased rapidly from August to November 2020 to a peak of 25% in North America (Table 1, Fig 4), and its dominant presence in the later USAFA samples may be due to stochastic events in the local community.

The pattern in ancestors A10, A11, A12 and A13 is strikingly different. For example, divergence from ancestor A3 was much lower, with an average of only 1.77 mutations/genome (s.d. 2.9, range 0–12). This frequency drops even further to < = 1 mutation/genome away from ancestors A11, A12 and A13 in the Colorado Springs variant lineage (Fig 3, S8 Table). Five mutations on the branch led to A10, and seven mutations on each of the branches led to two of its descendants, A11 and A12 (Fig 3). Furthermore, the highly derived A12 gave rise to 14 descendant sequences in the sample, including another shared ancestor differing by two mutations (A13). All of the descendant sequences from A12 and A13 differ from their ancestors by zero to three mutations, and there are respectively three and four sequences corresponding exactly to A13 and A14. The ancestors A10 and A11 have a few descendant sequences from CU individuals, indicating that they and A12 may have arisen from community spread. In contrast, the high prevalence of closely related USAFA samples in A12 and its descendants indicates sequence documentation of a rapid-spreading event, possibly involving one or more individuals.

We further validated this inference by estimating the divergence times between the SARS-CoV-2 samples collected by the USAFA (Fig 5) [30–32]. The node where the CO Springs variant branches off from the ancestral lineage was found to have a posterior distribution with a mean of twenty-four days before the most recent sample, November 19, 2020 (Fig 5). This puts the mean, estimated date of divergence at October 26[th], 2020, with a credible range (95%

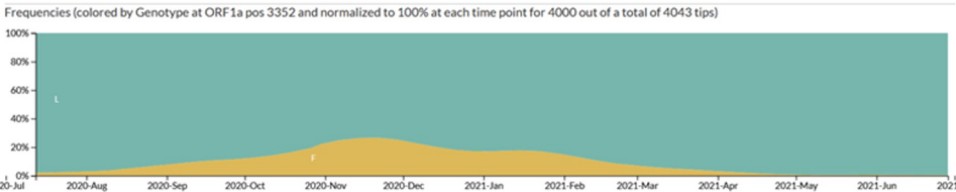

**Fig 4. Frequency of a CUAF clade 20G mutation in North America between March 2020 and June 2021.** The mutation pictured here is from ORF1A, genomic position 10319, amino acid position 3352.

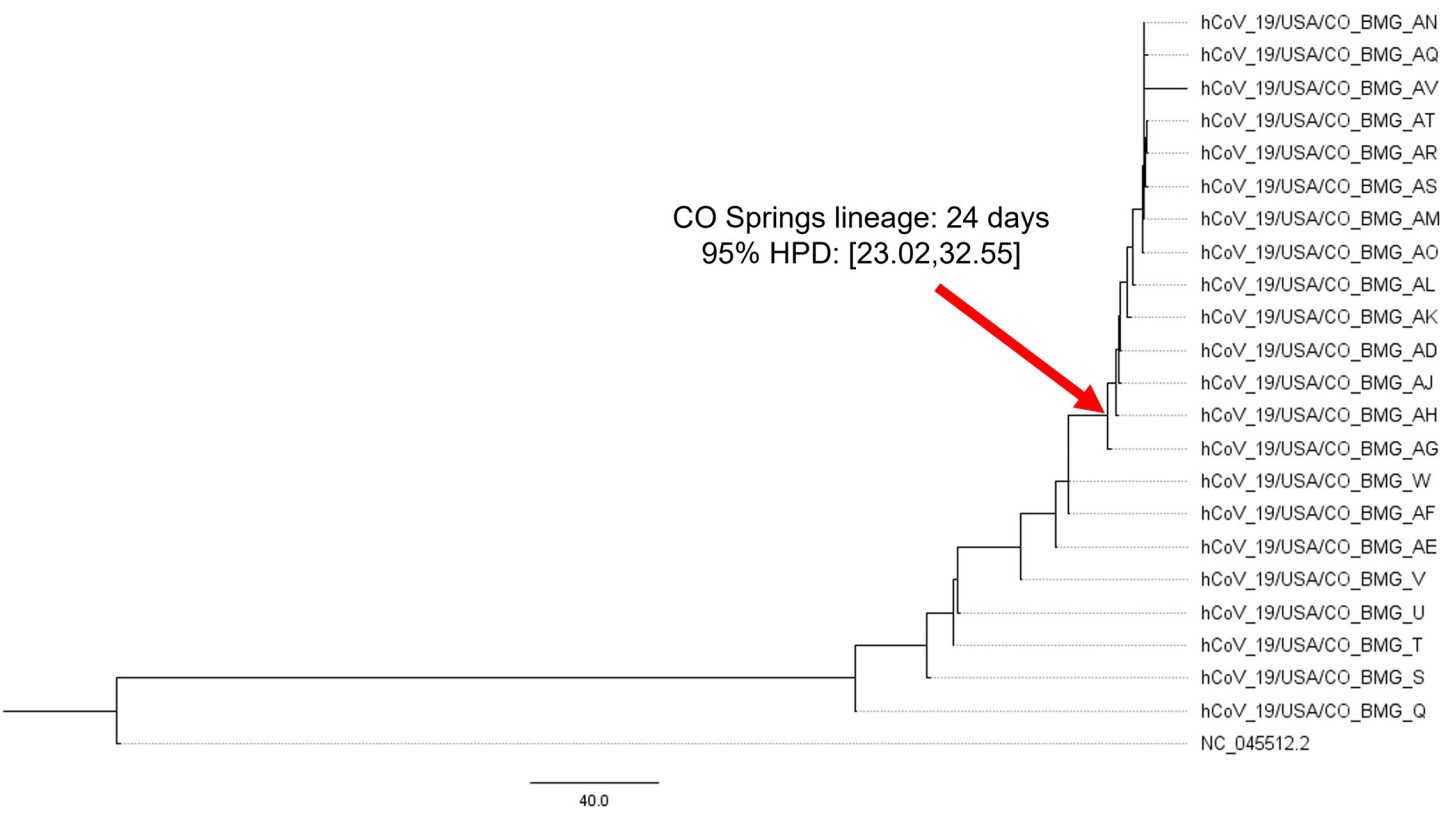

**Fig 5. Divergence estimates indicate that the CO Springs lineage originated around October 26th, 2020.** The red arrow indicates the node where the CO Springs lineage branches off from ancestral nodes. 95% HPD and tree distances are in units of "days before November 19, 2020".

HPD) between October 27, 2020 and October 17th, 2020. This range of credible divergence estimates is in accordance with USAFA records that trace the origin of the rapid spreading event to late October 2020 (SDataset 1).

Given the possibility that the Colorado Springs variant clade had some transmission advantage, we next considered the potential for these mutations to affect protein function. We evaluated the set of nine mutations that defined the CO Springs Variant ancestral clades A12–A13 (Table 2). These mutations have population frequencies < = 1% in NextStrain (Table 1), and three result in non-synonymous amino acid substitutions. Two of these mutations (ORF1A/

**Table 2. Lineage-defining mutations contributing to the Colorado Springs variant [A12-A13].**

| Ancestor | Mutation position | ORF | Protein | REF allele | ALT allele | Amino acid position | Codon change | Amino acid change | Biochem property change? |
|---|---|---|---|---|---|---|---|---|---|
| A13 | 829 | ORF1A | Nsp2 | C | T | 188 | AAC->AAT | N->N | N |
| A12 | 4021 | ORF1A | Macro Domain | C | T | 1252 | AAC->AAT | N->N | N |
| A12 | 7086 | ORF1A | Nsp3_C | C | T | 2274 | ACT->ATT | T-> I | Y |
| A12 | 12295 | ORF1A | Nsp8 | C | T | 4010 | ACT->ACC | T->T | N |
| A12 | 13216 | ORF1A | Nsp10 | T | C | 4317 | GAT->GAC | D->D | N |
| A12 | 14187 | ORF1B | RdRp | G | A | 250 | AGG->AGA | R->R | N |
| A12 | 21390 | ORF1B | Methyltransferase | A | G | 244 | TTA->TTG | L->L | N |
| A13 | 21830 | S | Spike | G | T | 268 | GTT->TTT | V->F | Y |
| A12 | 25593 | ORF3a | Orf3a | G | C | 67 | AAG->AAC | K->N | Y |

T2274I and ORF3A/K67N) arose on the branch leading to A12, the ancestor of the Colorado Springs variant. Based on literature review (see discussion), it appears likely that all three mutations impact the biochemical properties of their associated protein and may have implications for viral fitness.

## Genomic surveillance methods complement high-level contact tracing

From August 2020 through December 2020, USAFA utilized a random surveillance testing program, where a percentage of cadets [4–15%] were tested daily to identify asymptomatic or mildly symptomatic SARS-CoV-2 cases. All SARS-CoV-2 positive patients were interviewed, close-contacts identified, and class schedules reviewed to assess for additional contacts. These individuals were then placed into quarantine with testing and monitoring before release. If the individuals identified during contact tracing were not cadets, these close contacts were contacted following local public health guidance.

This method allowed for the identification of infection from a student, sports team, or community exposure. We were able to link multiple infections back to a rapid spreading event in late October 2021, which resulted in the strain clusters A12 and A13. Following the dramatic increase in infections, further lockdown measures were implemented at USAFA limiting new community introductions. At the end of the semester, a similarly rigorous testing and quarantine process occurred prior to release for the winter break, likely eliminating the strain from circulation within the population. The ability to contextualize genomic data with contact tracing information helps see a clearer picture for strain introduction, mutation, and propagation, while making assessment of subsequent viral fitness as SARS-CoV-2 continues to change.

## Discussion

Our robust sequencing provides a snapshot of infections in Colorado in late summer and early fall 2020. There were many circulating variants in Colorado at this time, and their dynamics broadly reflect strain variation and divergence across the rest of the US. In addition, our analysis suggests that there were likely multiple introductions of SARS-CoV-2 into the USAFA cadet population, despite their restricted interaction with Colorado Springs. Most did not lead to subsequent outbreaks, and only one sustained a rapid evolutionary expansion, which we name the Colorado Springs variant. Prior sequences from the nearby Denver/Aurora area and pre-outbreak USAFA samples (likely reflecting the Colorado Spring community) indicate slow and potentially neutral evolution of variant twigs, which come from common ancestors of known expanding variants previously identified by NextStrain. The lineages immediately prior and adjacent to the Colorado Springs variant, in contrast, indicate bursts of evolution including amino acid altering mutations that may have affected its transmission properties. This variant may have been highly contagious, but its spread also appears to have been promoted by one or more rapid spreading events. Luckily, it appears to have been contained by the rigorous epidemiological control procedures (e.g. social distancing, mask wear, limited gathering, virtual learning) employed at the USAFA. The type of focused community-level data collected here may be key to understanding how SARS-CoV-2 spreads in local settings, which may often have highly idiosyncratic dynamics compared to the country as a whole.

We employed rigorous sequencing quality control and validation steps, including standard PCR and sequencing replicates of all samples, further replicates of any moderately ambiguous results, and comparison to evolving ancestral sequences as well as the standard Wuhan reference. This ultimately resulted in an inferred ancestral sequence network that contained only two convergent mutations and was parsimonious. We verified our final ancestral network via maximum likelihood phylogenetic tree construction. Our results concerning the USAFA

diversification are not dependent on the fine details of the network, but it seems clear that more quantitative studies might be misled if they did not take a similarly cautious and rigorous replication approach with hands-on evaluation of the data, as we did here. Because we identified discrepancies that would have been accepted in less thorough [e.g., single-sequence replicate and single reference variant identification] protocols, we suspect, as do others, that an unknown number of variants in the GISAID database contain flawed sequences that may mislead phylogenetic and convergence analyses [23, 26, 41, 42].

The sequencing initiative presented here highlights the power of robust genomic surveillance to describe local viral dynamics, particularly when paired with epidemiological data collected with the patient samples. Sitko et al. collected these samples through a highly effective COVID-19 monitoring system at USAFA. Their approach relied on random sampling of individuals regardless of symptoms. In addition to their robust contact tracing steps, the USAFA was able to pinpoint their surge to a rapid spreading event on their campus in late October. This epidemiological conclusion agrees with the results described here, where we were able to phylogenetically reconstruct the divergence of these samples over the period of time from August to late November. Adopting this approach enabled the identification of the Colorado Springs variant in the context of enough branches to pinpoint a burst of change leading to the variant, and documented the spread of the variant to a large number of people over a short period of time. Through the use of evolutionary modelling, we traced this burst back to a date between October 17–27, 2020, with the highest probability of having occurred on October 26[th], 2020. It is known that immunocompromised individuals can serve as accelerated cauldrons of intra-host viral evolution with selected and rapid accumulation of epistatically interacting mutations [25], which might be an explanation for the burst of evolution we see here leading to the CO Springs variant. However, we do not know of such a case in the community, and the USAFA population is mostly young and extremely healthy, and the cadets likely interact with similarly young and healthy individuals in the local community. The possibility that rapid intra-host evolution could occur in such individuals, perhaps during long-term but largely asymptomatic infections, warrants consideration for further study.

It is important to track and model the evolution of highly adaptive strains that tend to rapidly rise in frequency in the population once they gain a sufficient foothold, but it is also important to describe patterns of viral evolution that may lead to attenuation. Such strains are likely to be found in sampling from asymptomatic patients because they tend to be less phenotypically severe cases. Attenuated strains have the potential to out-compete more severe strains due to the trade-off between virus transmissibility and severity [43]. Studies like this are well suited to capture a snapshot of this kind of variance, as samples were collected from both symptomatic and asymptomatic individuals, per USAFA's randomized testing surveillance protocol [15]. Further, mutations in these strains can create reservoirs of mostly neutral mutations, possibly leading to gradual genetic drift over time [44]. In the event of a transmission bottleneck, variants could then rise to sustained, high frequency [44, 45]. Such a scenario could contribute to antigenic shifts and viruses with a capacity to reduce vaccine efficacy [43]. These scenarios, in which neutral variants propagate by chance, seem plausible as the default mode of spreading for SARS-CoV-2 [46], punctuated by the rise of more transmissible variants of concern. While the Colorado Springs variant appears to have been confined to the isolated context in which it was found, similar variants may not be contained, and thus it is important to characterize them whenever possible.

The ancestor of the Colorado Springs variant (A12) contained two intriguing non-synonymous amino acid substitutions: ORF1A/T2274I and ORF3A/K67N. ORF1A/T2274I results in a shift from a polar uncharged to a non-polar residue at the third position of a three residue N-linked glycosylation site in the Nsp3 peptide. This type of post-translational modification to

Nsp3 is thought to be important for insertion into the endoplasmic reticulum of host cells, though it is not known how a mutation at a glycosylation site would impact its ability to do so [47]. However, disruption of N-linked glycosylation on other viral peptides has been shown to be destabilizing and negatively impacts virus viability [47, 48]. The ORF3A/K67N mutation results in a change from a positively charged residue to an uncharged, polar residue. This particular residue occurs in an LKK peptide motif that is predicted to be a likely B-cell epitope by Azad and Khan, 2021. Because the mutation seems to increase the free energy of folding, it has the potential to alter a putative B-cell epitope, allowing the virus to better evade host immune responses [49]. Interestingly, the only recorded North American SARS-CoV-2 genomes containing this mutation are found in Mexico (S6 Fig). The NextStrain database (derived from the GISAID database) is a highly incomplete collection of existing viral strains, making it difficult to determine whether this mutation was imported to the Colorado area or arose *de novo* locally. The third amino acid altering mutation, along the lineage leading to A13, is located in the Spike protein at position 268. The change converts a small, hydrophobic valine residue to phenylalanine, which is also hydrophobic but contains a large six-carbon ring side chain. A shift in steric properties is likely to impact local structure, and thereby potentially modify protein function. Without further experimental studies, it is difficult to know how these mutations affect viral dynamics and the extent to which they enabled the rapid spreading event.

This case study, while limited in size and scope, is an exemplar to describe the viral phylodynamics of a locally confined rapidly-spreading transmission event, in combination with paired epidemiological data. Due to their rapid expansion, coupled with minimal mutation accumulation, rapid spread scenarios have little phylogenetic structure to describe [50], and the contact structures involved may strongly deviate from the average assumptions used in most epidemiological models [14, 51]. We conclude that case studies similar to that presented here could assist in outbreak control, provide variant-origin replicates to obtain a broader view of the process and refine epidemiological models, and help in early detection and action against novel variants of concern when they occur in the future.

## Supporting information

**S1 Fig. Average matching amplicon position of sequencing primers.** 1A shows the average position along the y axis, primer number along the x axis. 1B shows the variance calculated for the distribution of matching positions for each primer.
(TIF)

**S2 Fig. Next Strain estimates of North American continental frequency of the D614G mutation, from March 2020-May 2021.** Gold indicates frequency of 'G' substitution over time, green indicates frequency of 'D' substitution over time.
(TIF)

**S3 Fig. Eleven major SARS-CoV-2 clades in North America between March 2020 and June 2021, across 3923 genomes.** A) Phylogenetic relationships of strains across the continent. B) Strain relative prevalence between August 2020 and July 2021. Date accessed July 12, 2021.
(TIF)

**S4 Fig. Depiction of the ancestral network of 44 Colorado SARS-CoV-2 genomes.** Each tip indicates the genome letter identifier and the number of mutations away from its most recent ancestral node. Ancestral nodes contain the name of the node (A1-A13) and the position of lineage-defining mutations which are inherited by all downstream lineages. Node and tip coloring described in legend.
(TIF)

**S5 Fig. Phylogenetic tree of 44 novel SARS-CoV-2 genomes from Colorado and Wuhan-Hu-1.** Genomes were aligned with Mugsy and the consensus tree was generated under the Jukes-Cantor model using IQTree. Tip labels indicate the sample names as identifiable in GISAID. Red branch labels indicate the major phylogroups described in this paper. Blue asterisk indicates the Wuhan-Hu-1 genome outgroup, NC_045512.2.
(TIF)

**S6 Fig. Geographic location of the Orf3a K67N mutation in NextStrain.** The only other documented instance of this variant in the NextStrain repository occurs in Northern Mexico. Map image downloaded from NextStrain, which uses OpenStreetMap®. OpenStreetMap® is open data, licensed under the Open Data Commons Open Database License (ODbL) by the OpenStreetMap Foundation (OSMF).
(TIF)

**S1 File. GISAID accession IDs for the 44 SARS-CoV-2 genomes sequenced in this work.**
(DOCX)

**S1 Table. List of successfully sequenced genomes.** Sample ID#s represent deidentified IDs.
(XLSX)

**S2 Table. Sequencing meta data and de-identified sampleIDs for successfully sequenced replicates.**
(XLSX)

**S3 Table. Variant calls that were excluded (yellow) during QC processing.**
(XLSX)

**S4 Table. Discordant genomes containing putative false positive variant calls that were either verified as true positive (green) or excluded as false positives (yellow).**
(XLSX)

**S5 Table. Genomes with low quality variants that were initially excluded (false positives) due to low quality, but were re-included upon phylogenetic re-assessment (green).**
(XLSX)

**S6 Table. Genomes containing convergent variants (blue).**
(XLSX)

**S7 Table. Non-lineage defining genome mutations.**
(XLSX)

**S8 Table. Average number of observed variants, by phylogroupings.**
(XLSX)

**S1 Dataset. Evolutionary model parameters, divergence estimates and MCMC data.**
(ZIP)

## Acknowledgments

We thank members of the Rissland and Pollock labs for helpful discussions.

## Author Contributions

**Conceptualization:** Kristen J. Wade, David D. Pollock, Olivia S. Rissland.

**Data curation:** Kristen J. Wade, Samantha Tisa, Chloe Barrington, Jesslyn C. Henriksen, Kristy R. Crooks, Christopher R. Gignoux, Austin T. Almand, J. Jordan Steel, John C. Sitko, Joseph W. Rohrer, Douglas P. Wickert, Erin A. Almand, David D. Pollock, Olivia S. Rissland.

**Formal analysis:** Kristen J. Wade, Samantha Tisa, Chloe Barrington, Jesslyn C. Henriksen, Kristy R. Crooks, Austin T. Almand, J. Jordan Steel, John C. Sitko, Joseph W. Rohrer, Douglas P. Wickert, Erin A. Almand, David D. Pollock, Olivia S. Rissland.

**Funding acquisition:** David D. Pollock, Olivia S. Rissland.

**Investigation:** Kristen J. Wade, Samantha Tisa, Chloe Barrington, Austin T. Almand, David D. Pollock, Olivia S. Rissland.

**Methodology:** Kristen J. Wade, David D. Pollock, Olivia S. Rissland.

**Project administration:** Kristen J. Wade, Samantha Tisa, Chloe Barrington, Kristy R. Crooks, Austin T. Almand, J. Jordan Steel, John C. Sitko, Joseph W. Rohrer, Douglas P. Wickert, Erin A. Almand, David D. Pollock, Olivia S. Rissland.

**Resources:** Kristen J. Wade, Christopher R. Gignoux, David D. Pollock, Olivia S. Rissland.

**Software:** Kristen J. Wade, David D. Pollock, Olivia S. Rissland.

**Supervision:** Kristen J. Wade, Chloe Barrington, Jesslyn C. Henriksen, Christopher R. Gignoux, Austin T. Almand, J. Jordan Steel, John C. Sitko, Joseph W. Rohrer, Douglas P. Wickert, Erin A. Almand, David D. Pollock, Olivia S. Rissland.

**Validation:** Kristen J. Wade, Samantha Tisa, Chloe Barrington, David D. Pollock, Olivia S. Rissland.

**Visualization:** Kristen J. Wade, David D. Pollock, Olivia S. Rissland.

**Writing – original draft:** Kristen J. Wade, Chloe Barrington, David D. Pollock, Olivia S. Rissland.

**Writing – review & editing:** Kristen J. Wade, Samantha Tisa, Chloe Barrington, Jesslyn C. Henriksen, Kristy R. Crooks, Christopher R. Gignoux, Austin T. Almand, J. Jordan Steel, John C. Sitko, Joseph W. Rohrer, Douglas P. Wickert, Erin A. Almand, David D. Pollock, Olivia S. Rissland.

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
