## [Decision Letter · Decision Letter 0]

16 Jun 2022

PONE-D-22-08994Phylodynamics of a regional SARS-CoV-2 rapid spreading event in Colorado in late 2020PLOS ONE

Dear Dr. Wade,

Thank you for submitting your manuscript to PLOS ONE. After careful consideration, we feel that it has merit but does not fully meet PLOS ONE’s publication criteria as it currently stands. Therefore, we invite you to submit a revised version of the manuscript that addresses the points raised during the review process.

Please perform a molecular dating and the estimation of the evolutionary rate. 

We look forward to receiving your revised manuscript.

Kind regards,

Etsuro Ito

Academic Editor

PLOS ONE

Journal Requirements:

"KW is supported by NIH R01 GM083127. CB is supported by the T32 grant T32GM136444 awarded to the Molecular Biology graduate program. This work was supported by NIH grants R35GM128680 (OSR) and the RNA Bioscience Initiative."

Reviewers' comments:

Reviewer's Responses to Questions

**Comments to the Author**

1. Is the manuscript technically sound, and do the data support the conclusions?

Reviewer #1: Yes

2. Has the statistical analysis been performed appropriately and rigorously? 

Reviewer #1: Yes

3. Have the authors made all data underlying the findings in their manuscript fully available?

Reviewer #1: Yes

4. Is the manuscript presented in an intelligible fashion and written in standard English?

Reviewer #1: Yes

5. Review Comments to the Author

Reviewer #1: The manuscript by Wade and colleagues is a very interesting piece of work.

Although I am not English native speaker, to my opinion the manuscript is well written in a clear language.

It describes the phylodynamics of a given lineage that to the Authors likely originated in the local Colorado Springs community and expanded rapidly over the course of two months in an outbreak within the well-controlled environment of the United States Air Force Academy.

Anyhow, the work-flow of analyses contains a big lacking: the molecular dating. I suggest the Author to perform a molecular dating and the estimation of the evolutionary rate. This kind of result will be very useful in order to have a much more complete point of view on the temporally origin and diffusion of the lineage.

6. PLOS authors have the option to publish the peer review history of their article (what does this mean?). If published, this will include your full peer review and any attached files.

Reviewer #1: No

---

## [Author Response · Author response to Decision Letter 0]

18 Aug 2022

We thank the reviewer for their thoughtful comments about our manuscript, and as we detail below, we have made changes to the manuscript to address the molecular dating concerns raised by the reviewer. 

Editor comments:

The manuscript style formatting has been revised according to the guidelines.

Thank you for bringing this matter to our attention; this statement has now been included.

The ethics statement has been moved to the Methods section.

The captions for Supporting Information are present in the text and at the end of the manuscript (after References), according to the SI guidelines. 

Reviewer comments:

Reviewer #1: The manuscript by Wade and colleagues is a very interesting piece of work.

Although I am not English native speaker, to my opinion the manuscript is well written in a clear language.

We thank the reviewer for their positive comments. 

It describes the phylodynamics of a given lineage that to the Authors likely originated in the local Colorado Springs community and expanded rapidly over the course of two months in an outbreak within the well-controlled environment of the United States Air Force Academy.

Anyhow, the work-flow of analyses contains a big lacking: the molecular dating. I suggest the Author to perform a molecular dating and the estimation of the evolutionary rate. This kind of result will be very useful in order to have a much more complete point of view on the temporally origin and diffusion of the lineage.

We have now performed Bayesian evolutionary modelling of the USAFA SARS-CoV-2 genomes, using their sampling dates from August-November 2020 to inform construction of the phylogenetic tree estimates and inference of divergence rates. This was performed using the Bayesian Evolutionary Analysis Sampling Trees (BEAST) software analysis suite. In doing so, we estimated the divergence of the novel CO Springs variant to a likely origin around October 26, 2020. This agrees with the predictions of the USAFA epidemiological modelling (Sitko et al., 2021), which traced the rapid spreading outbreak back to an event in late October 2020. Our additional modelling and divergence estimate provides further resolution of the related epidemiological data and more robustly supports the conclusions of this manuscript. This new analysis has been included in the revised manuscript.

---

## [Decision Letter · Decision Letter 1]

22 Aug 2022

Phylodynamics of a regional SARS-CoV-2 rapid spreading event in Colorado in late 2020

PONE-D-22-08994R1

Dear Dr. Wade,

We’re pleased to inform you that your manuscript has been judged scientifically suitable for publication and will be formally accepted for publication once it meets all outstanding technical requirements.

Kind regards,

Etsuro Ito

Academic Editor

PLOS ONE

Reviewers' comments:

Reviewer's Responses to Questions

**Comments to the Author**

1. If the authors have adequately addressed your comments raised in a previous round of review and you feel that this manuscript is now acceptable for publication, you may indicate that here to bypass the “Comments to the Author” section, enter your conflict of interest statement in the “Confidential to Editor” section, and submit your "Accept" recommendation.

Reviewer #1: All comments have been addressed

2. Is the manuscript technically sound, and do the data support the conclusions?

Reviewer #1: Yes

3. Has the statistical analysis been performed appropriately and rigorously? 

Reviewer #1: Yes

4. Have the authors made all data underlying the findings in their manuscript fully available?

Reviewer #1: Yes

5. Is the manuscript presented in an intelligible fashion and written in standard English?

Reviewer #1: Yes

6. Review Comments to the Author

Reviewer #1: I am fully satisfied by the changes made by the authors.

To my opinion the manuscript can be published in this version as it is.

7. PLOS authors have the option to publish the peer review history of their article (what does this mean?). If published, this will include your full peer review and any attached files.

Reviewer #1: No

---

## [Editor Report · Acceptance letter]

26 Sep 2022

PONE-D-22-08994R1 

Phylodynamics of a regional SARS-CoV-2 rapid spreading event in Colorado in late 2020 

Dear Dr. Wade:

I'm pleased to inform you that your manuscript has been deemed suitable for publication in PLOS ONE. Congratulations! Your manuscript is now with our production department. 

Kind regards, 

on behalf of

Prof. Etsuro Ito 

Academic Editor

PLOS ONE